

# Psychometric properties of the Mexican version of the opening minds stigma scale for health care providers (OMS-HC)

Hugo Noel Valdivia Ramos[1], Jazmín Mora-Rios[2], Guillermina Natera[2] and Liliana Mondragón[2]

[1] Programa de Maestría y Doctorado de Ciencias Médicas, Ontológicas y de la Salud, Universidad Nacional Autónoma de México, Ciudad de México, Mexico
[2] Dirección de Investigaciones Epidemiológicas y Psicosociales, Instituto Nacional de Psiquiatría Ramón de la Fuente Muñiz, Ciudad de México, México

Corresponding author
Jazmín Mora-Rios,
morarj@inprf.org.mx

## ABSTRACT

**Background:** Healthcare providers are one of the main groups that contribute to the stigmatization of people with mental disorders. Apathy, accusation, fatalism, and morbid curiosity are the most common forms of stigmatization encountered, and these are associated with inadequate treatment, reduced treatment adherence, decreased help-seeking behavior, an increased risk of relapse, and complications with other medical conditions. The aim of this study was to examine the psychometric properties of an adapted Spanish version of the Opening Minds Stigma Scale (OMS-HC) for healthcare providers in Mexico and identify certain stigmatizing attitudes within this group.

**Methods:** An ex-post facto cross-sectional observational study was conducted with 556 healthcare providers in Mexico, with an average age of 29.7 years, who were mostly women (80.4%). Validity was examined through confirmatory factor analysis. Differences according to gender, discipline, occupation, and educational level were analyzed using multivariate methods.

**Results:** The factor structure of the OMS-HC, consisting of three subscales identified by the original authors of the instrument (attitudes of healthcare providers towards people with mental illness, secrecy/help-seeking, and social distance), was confirmed. The model demonstrated good fit ($x^2/df$ = 2.36, RMSEA = 0.050, CFI = 0.970, TLI = 0.962, SRMR = 0.054, NFI = 0.950, PNFI = 0.742). Internal consistency was found to be adequate ($\alpha$ = 0.73, $\omega$ = 0.76) for the scale itself and slightly lower than acceptable for the subscales. Significant differences were found by discipline, educational level, and, for student providers, by academic semester. Higher scores were observed on the OMS-HC scale among nursing and medical professionals, undergraduate students, and those in early semesters.

**Conclusions:** The Spanish version of the OMS-HC has demonstrated adequate psychometric properties and could be a useful tool to facilitate research on this topic in Mexico, and to carry out comparative studies with healthcare personnel in other Spanish-speaking countries.

## INTRODUCTION

Mental health disorders and substance abuse are currently one of the leading causes of disability worldwide, accounting for 13% of the global disease burden (*World Health Organization, 2021*). In the case of Mexico, they represent 16% of all disability-adjusted life years (DALYs) and 33.5% of all years lived with disability (YLDs) (*Pan American Health Organization, 2018*). The stigma associated with mental health disorders impedes timely and effective care for individuals. This issue is particularly relevant in low- and middle-income countries where there is less research and attention than in high-income countries (*Wainberg et al., 2017*).

Stigma is defined as the co-occurrence of labeling, stereotyping, separation, loss of status, and discrimination in a situation where power is exercised (*Link & Phelan, 2001*). In addition to coping with their condition, those with mental health disorders are forced to deal with misinformation on the part of society and being the object of prejudice and rejection, which affects their well-being and quality of life (*Martínez & Hishaw, 2016*).

Stigmatizing attitudes toward individuals with mental health disorders have been identified in healthcare providers in various healthcare services, including specialized ones. These attitudes take on various forms, such as mockery, indifference, blame, fatalism, and morbid curiosity. Unfortunately, these negative attitudes can lead to poor care, treatment non-adherence, increased risk of relapse, and other medical complications (*Livingston & Boyd, 2010*; *Dubreucq, Plasse & Franck, 2021*).

In recent decades, there has been a growing interest in reducing these attitudes among healthcare providers, to combat discriminatory practices and improve medical care for people with mental health disorders (*Griffiths et al., 2014*). The scientific literature has documented various stigmatizing attitudes in healthcare providers and associations with their age, educational level, and work experience (*Mora-Ríos, Ortega-Ortega & Natera, 2016*; *Rivera-Segarra, Varas-Díaz & Santos-Figueroa, 2019*). Although some studies have disagreed over these results (*Kopera et al., 2015*; *Carrara et al., 2019*), others have found that increased contact with people with mental health disorders can reduce stigmatizing attitudes (*Griffiths et al., 2014*; *Stuber et al., 2014*). These findings suggest that technical knowledge and skills alone may not be enough to achieve behavior change among healthcare providers (*Schulze, 2007*).

In Latin America, research on stigma is much more limited than in high-income countries (*Semrau et al., 2015*). It is estimated that 7.1 million people in Mexico live with some type of disability, of which 19.6% are related to emotional or mental problems (*Instituto Nacional de Estadística y Geografía, 2017*), and of these, only a minority have access to specialized services. This gap in care is as high as 80% in low- and middle-income countries (*Gómez-Dantés & Frenk, 2018*), and studies have shown that stigma is one of the major barriers in countries such as Mexico and Brazil (*Andrade et al., 2014*). Ignorance and shame regarding these problems prevail not only in the general population, but also among healthcare personnel. One study of 59 persons with a psychiatric diagnosis found that healthcare personnel were the second-most important source of their stigmatization (*Mora-Ríos, Ortega-Ortega & Natera, 2016*).

It is necessary to have instruments for the evaluation of stigma that not only have adequate psychometric properties, but that are also adapted to the cultural context of the population (*Yang et al., 2014*). One study based on a systematic review of stigma measures found that some of these, besides having a large number of items, showed distinct problems of validity (*Sastre-Rus et al., 2019*); the challenge is to develop measurements that meet adequate levels of validity and reliability.

Instruments to evaluate stigma in Latin America are mainly for the general population (*Mora-Ríos et al., 2013*; *Mora-Ríos & Ortega-Ortega, 2021*). However, instruments specifically for the evaluation of healthcare personnel have been adapted and developed in recent years (*Sapag et al., 2019*; *Vielma-Aguilera et al., 2023*). These allow for the identification of stigmatizing attitudes in the healthcare environment, which is essential to improving the quality of care and promoting inclusion and respect toward people with mental health disorders.

The Opening Minds Stigma Scale for Health Care Providers (OMS-HC) was developed to assess the attitudes of healthcare providers toward mental illness (*Kassam et al., 2012*). The original version had a total of twenty items; this was adjusted to a two-factor structure with twelve items. However, a subsequent validation by *Modgill et al. (2014)* resulted in a three-factor version with fifteen items. This version of the instrument has demonstrated adequate psychometric properties, including good internal consistency both globally ($\alpha = 0.79$) and in the three subscales that make it up: (1) attitudes of healthcare providers toward those with mental illness ($\alpha = 0.68$), (2) secrecy/help-seeking ($\alpha = 0.67$), and (3) social distance ($\alpha = 0.69$). The OMS-HC scale has been widely adopted in international research (*Papish et al., 2013*; *Sastre-Rus et al., 2019*) and used to evaluate interventions in various populations, professional settings, and online educational programs (*Knaak, Ungar & Patten, 2015*; *Fernandez et al., 2016*; *Chang et al., 2017*).

The OMS-HC scale has Spanish-language adaptations for Spain and Chile. The version for Spain showed adequate fit indexes ($x^2/df = 1.348$, RMSEA = 0.065, CFI = 0.961, TLI = 0.920) (*Őri et al., 2023*), as did the Chilean version (RMSEA = 0.052, CFI = 0.832, TLI = 0.738, SRMR = 0.048) (*Sapag et al., 2019*). For both adaptations, the authors reported greater levels of reliability for the global scale than for the three original subscales.

The OMS-HC scale has some advantages over other measurements of stigma: it is an instrument specially designed for healthcare providers, it has adequate psychometric properties on the international level, and there are Spanish-language adapted versions. It can also incorporate variables related to social distance and secrecy/help-seeking that can be very useful in evaluating interventions to address stigma. Although there are already two Spanish-language versions of this scale, our interest, following the suggestions of *Yang et al. (2014)*, is in developing an instrument that is culturally adapted to the Mexican population. To do so, we employed a process of semantic validation of the original version of the scale, which will be useful for making comparisons with the factorial structure of the Spanish-language versions adapted for Chile and Spain.

The objective of this study is thus to carry out a factorial validation of the OMS-HC in Mexican healthcare personnel in order to evaluate the psychometric properties of the instrument. The study also analyzes differences in stigma levels as a function of

sociodemographic variables such as age, gender, educational level, occupation, discipline, and academic semester. A version of this scale adapted for the Mexican population could be useful in carrying out comparative studies with other Spanish-speaking countries.

## MATERIALS AND METHODS

### Study design

An ex post facto, cross-sectional observational study was designed. The research team established contact with four family medicine health clinics in Mexico City and three universities, which expressed their interest in participating in the study. Approval was then obtained from the participating institutions and dates were scheduled for administering the questionnaires. The institutions provided the necessary facilities to carry out the administration and allowed the invitation for voluntary participation by individuals from the fields of medicine, nursing, psychology, and social work.

Data collection began in February 2020, with responses from 143 participants. It was interrupted, however, by social distancing measures and limited access to health centers during the COVID-19 pandemic. A decision was made to continue with an online platform and necessary adjustments. Healthcare workers were invited to participate, with an assurance that participation would be voluntary. An informed consent statement was included, and the instructions and scale used were the same. Online data collection began in September 2020 and was completed in December of that year; it included 462 participants.

### Participants

Non-probabilistic convenience sampling was used. To determine the correct sample size, the authors used the recommendation of MacCallum et al. (1999) to obtain a sample of at least 500 participants to obtain an adequate factorial structure.

The inclusion criteria for participation in the study were the following: over 18 years of age, residing in Mexico City, and working in the healthcare field as either a student or a professional. There were a total of 605 participants in the sample, but analysis included only fully completed questionnaires, resulting in an effective sample of 556 participants (92%), whose sociodemographic characteristics are described in Table 1. The mean age of the participants was 29.7 years ($SD$ = 9.45), 80.4% ($n$ = 447) women and the remaining 19.6% men ($n$ = 109). Most of the participants came from the disciplines of medicine (59%, $n$ = 328) and nursing (20.3%, $n$ = 113), and 79.5% ($n$ = 442) held a bachelor's degree. Of the total, 44.2% ($n$ = 246) were students, 40.3% ($n$ = 224) were professionals, and 15.5% were both ($n$ = 332). For the latter category, academic semester was considered from the first undergraduate semester through the graduate program. Finally, 23.4% ($n$ = 78) were pursuing a specialty, although we did not collect data on their fields of study.

### Instruments

The original Opening Minds Scale for Health Care Providers (OMS-HC) was developed in English by Kassam et al. (2012) to evaluate the attitudes of healthcare providers toward individuals with mental disorders. It has a factorial structure consisting of two dimensions

**Table 1 Sociodemographic characteristics of the sample.**

| Characteristic | n = 556 | % |
|---|---|---|
| Age | | |
| (Years) | 18–72 | |
| (Mean) | 29.7 | |
| Gender | | |
| Female | 447 | 80.4 |
| Male | 109 | 19.6 |
| Discipline | | |
| Medicine | 328 | 59 |
| Nursing | 113 | 20.3 |
| Clinical psychology | 71 | 12.8 |
| Others | 44 | 7.9 |
| Educational level | | |
| Technical education | 23 | 4.1 |
| Bachelor's degree | 442 | 79.5 |
| Master's degree | 81 | 14.6 |
| Doctoral degree | 10 | 1.8 |
| Occupation | | |
| Student | 246 | 44.2 |
| Professional | 224 | 40.3 |
| Both | 86 | 15.5 |
| Academic semester[a] | | |
| 1°–4° semester | 17 | 5.1 |
| 5°–6° semester | 44 | 13.2 |
| 7°–8° semester | 58 | 17.4 |
| 9°–10° semester | 40 | 12 |
| Social service | 57 | 17.1 |
| Specialization | 78 | 23.4 |
| Graduate program | 38 | 11.4 |

**Note:**
[a] Only "student" and "both" categories were included (n = 332).

that account for 45% of the variance using twelve of the twenty proposed items. These dimensions include attitudes of healthcare providers toward mental illness ($\alpha = 0.75$) and attitudes of secrecy toward mental illness ($\alpha = 0.72$). The first dimension contains seven items, while the second contains five. The scale has adequate levels of global internal consistency ($\alpha = 0.82$) and an interclass correlation of 0.66 (95% CI [0.54–0.75]).

For this study, the 15-item version proposed by *Modgill et al. (2014)* was used, in which three dimensions are identified: (1) attitudes of healthcare providers toward people with mental illness, (2) secrecy/help-seeking, and (3) social distance, based on the factorial validation. The answer form includes a five-point Likert scale (completely agree, agree, neither agree nor disagree, disagree, and completely disagree). Higher scores on the scale indicate greater stigmatization. Items 2, 6, 7, 8, and 14 are reverse scored. A section on

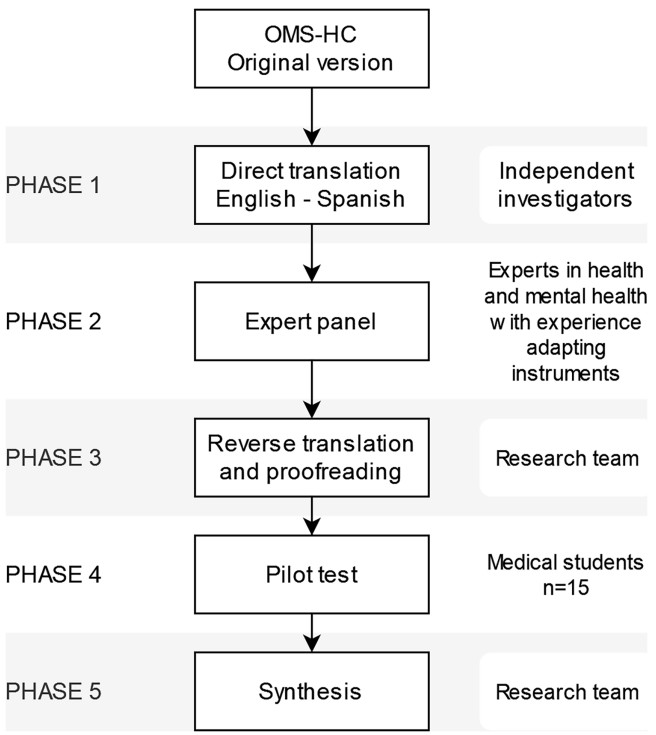

**Figure 1** Adaptation process of the OMS-HC scale to the Spanish version.

sociodemographic data was included, comprising age, gender, educational level, occupation, discipline, and, in the case of those undergoing training, the semester in which participants are were enrolled.

## Procedure

The authors have permission from the copyright holders, *Kassam et al. (2012)*, to use the OMS-HC instrument. The Spanish adaptation of the scale was developed using the rational criteria method with direct translation, which involves a consensus among experts who analyze the contents of the scale in the original language to ensure its correct translation. The instrument underwent adaptation in five main phases until a culturally relevant version was obtained for the study population (Fig. 1). In the first phase, which consisted of a direct translation from English to Spanish, a method of expert consensus was followed, according to the recommendations of *Sousa & Rojjanasrirat (2011)*.

Two bicultural and independent investigators collaborated in this stage, each with a background in different fields of mental health, one in psychiatry and the other in psychology. Two versions were thus obtained, which were then compared by a key informant with experience in mental health. This key informant resolved discrepancies between the two versions and provided the resulting version to a panel of experts. In the second phase, the panel of experts, consisting of three researchers with experience in mental health, evaluated the theoretical relevance, clarity, and appropriate language for the Mexican population. In the third phase, the research team carried out a reverse translation

of the instrument to verify the precision and quality of the translation. In the fourth phase, a pilot test was conducted with fifteen medical students who evaluated the clarity of the instructions and items using a dichotomous scale (clear or ambiguous). Finally, in the fifth phase, the observations from the pilot study were thoroughly reviewed. Errors of grammar, spelling, and style were corrected to insure that the final version was coherent and precisely reflected the original meaning of the items on the scale. The Spanish adaptation can be found in the supplementary information in this study (see File S1).

## Ethical considerations

The study was approved by the Ethics Committee of the National Autonomous University of Mexico (Approval No. Ext/01/2019). The study adhered to the ethical criteria established in the international ethical guidelines for biomedical research in human subjects (*Council for International Organizations of Medical Sciences, 2016*). The study entailed minimal risk and participation was voluntary. The informed consent form included an explanation of the objectives of the study, while ensuring confidentiality, privacy, and other ethical guarantees for participants.

## Data analysis

Descriptive statistics were used to analyze sociodemographic data. A confirmatory factor analysis (CFA) was performed to evaluate the factorial structure of the instrument. Before the CFA, the Kaiser-Meyer-Olkin (KMO) sampling adequacy index and Bartlett's assumption of sphericity were calculated. A parallel analysis was conducted to corroborate the factorial structure suggested by *Modgill et al. (2014)*. The model was subsequently calculated using a three-factor CFA using the weighted least squares estimator with adjusted mean and variance (WLSMV) (*Li, 2016*).

Multiple indicators were employed to assess the model's fit (*Hu & Bentler, 1999*; *Schermelleh-Engel, Moosbrugger & Müller, 2003*). The ratio of chi-squared to degrees of freedom ($x^2/df$) was used to measure the discrepancy between the data and the hypothesized model, with a result between one and three considered a good fit (*Schermelleh-Engel, Moosbrugger & Müller, 2003*). The root mean square error of approximation (RMSEA) was used as an index based on covariances; the model is acceptable if its value is less than 0.05 (*Hu & Bentler, 1999*). The comparative fit index (CFI) was used to contrast the loss produced by the change from the proposed model to the null model, in which a value equal to or greater than 0.95 is deemed optimal (*Hu & Bentler, 1999*). The Tucker-Lewis Index (TLI) was used to indicate the proportion of total information explained by the model, and a value equal to or greater than 0.95 was considered a good level of fit (*Schermelleh-Engel, Moosbrugger & Müller, 2003*). The normalized fit index (NFI) was utilized to indicate the proportion of variance and covariance explained by the model compared to the null model, with values close to one being considered a good level of fit. The standardized root mean square residual (SRMR) was included, and a value below 0.08 was considered a good fit (*Schermelleh-Engel, Moosbrugger & Müller, 2003*). The parsimony normed fit index (PNFI) was used to evaluate the relationship between the constructs and the theory, and a model was deemed

to have a good fit if the value was greater than 0.60, with better values closer to one (*Mulaik et al., 1989*).

The internal consistency of the instrument, both overall and by subscale, was assessed using McDonald's omega and Cronbach's alpha coefficients (*McDonald, 1999*; *Tavakol & Dennick, 2011*). Reliability values equal to or greater than 0.70 for both coefficients are considered acceptable (*McNeish, 2017*). The means of the OMS-HC were calculated and compared with the sociodemographic data using Student's $t$-tests and an ANOVA, with Tukey's test utilized as a *post-hoc* analysis. Before analysis, data homogeneity was assessed by Levene's test. Mann-Whitney and Kruskal-Wallis $U$-tests were performed as nonparametric analysis to confirm results. The relationship between quantitative variables was analyzed using Spearman's rho. All analyses were performed using R statistical software version 4.0.3 (*R Core Team, 2016*) and G*Power software version 3.1.9.7 (*Erdfelder, Faul & Buchner, 1996*).

# RESULTS

A total of 556 participants who completed all items on the OMS-HC questionnaire were included in the sample for analysis. The data showed a satisfactory sample adequacy measure (KMO) of 0.782, as well as a significant Bartlett sphericity test with $df = 105$, suggesting that the data was suitable for factor analysis. Further analysis, using parallel analysis, identified the presence of three common factors.

To confirm the appropriateness of the three-factor model for this sample, confirmatory factor analysis (CFA) was performed. The three-factor model demonstrated consistency with the proposed theoretical model and showed good fit indicators. Specifically, the ratio of chi-squared to degrees of freedom was 2.36 (193.765/82), the root mean square error of approximation (RMSEA) was 0.050, the comparative fit index (CFI) was 0.970, the Tucker-Lewis index (TLI) was 0.962, the normalized fit index (NFI) was 0.95, the standardized root mean square residual (SRMR) was 0.054, and the parsimony normed fit index (PNFI) was 0.742.

All the standardized loads of the items were higher than the criterion of 0.3, indicating that the items were well-represented by their respective factors. Additionally, the covariances by factor indicated correlation among the three subscales. The final solution of the model is presented in Fig. 2.

Table 2 presents the results of the internal consistency analysis and item correlation of the OMS-HC scale. Corrected correlation values between each item and the total questionnaire score ranged from 0.23 to 0.57, with all items showing a corrected correlation above 0.2. Cronbach's alpha values if each item was removed did not indicate significant changes in the global value of the scale. The global internal consistency of the scale was adequate with an alpha value of 0.73 and a McDonald's omega of 0.76. Cronbach's alpha values were 0.61, 0.60, and 0.51 for healthcare providers' attitudes toward people with mental illness, social distance, and secrecy/help-seeking, and McDonald's omega for the subscales were 0.70, 0.63, and 0.61, respectively.

Table 3 shows the means, standard deviations, and distribution of responses for the three subscales and their respective items. As can be seen, the majority of participants had

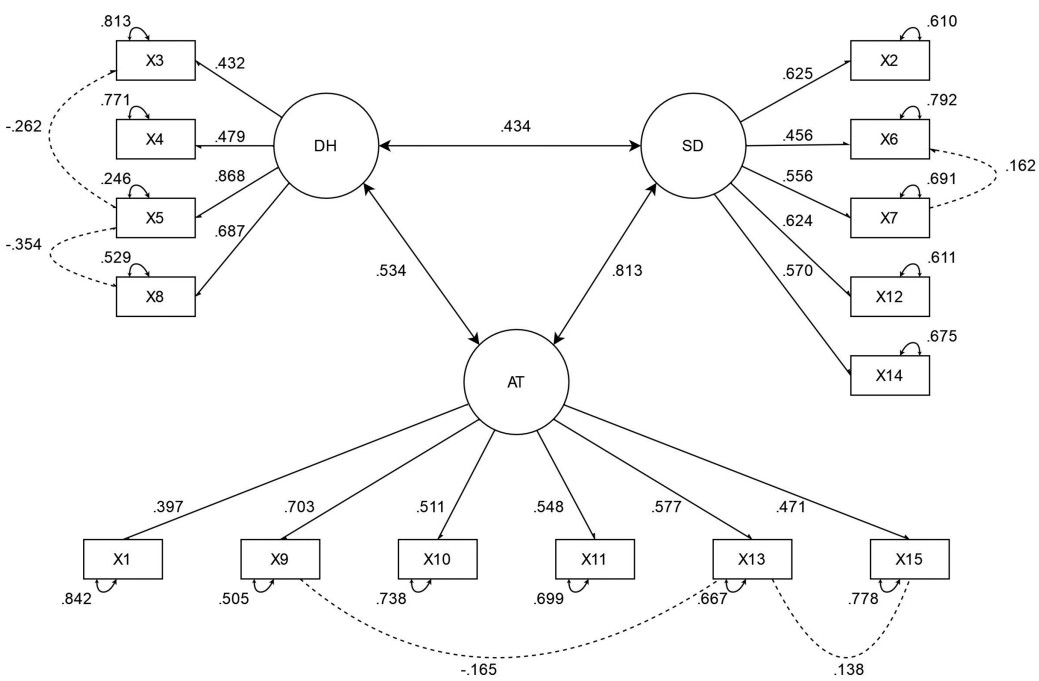

**Figure 2 Factorial solution of the OMS-HC.** AT, Attitudes of health personnel toward people with mental illness; DH, disclosure/help-seeking; SD, social distance. Item numbers refer to the version adapted by *Modgill et al. (2014)*.               

**Table 2 Internal consistency analysis and item correlation of the OMS-HC scale.**

| Item number | Corrected item-total correlation | Cronbach's alpha if item deleted |
|---|---|---|
| Item 1 | 0.35 | 0.72 |
| Item 2i | 0.41 | 0.72 |
| Item 3 | 0.23 | 0.73 |
| Item 4 | 0.34 | 0.72 |
| Item 5 | 0.37 | 0.72 |
| Item 6i | 0.31 | 0.73 |
| Item 7i | 0.48 | 0.71 |
| Item 8i | 0.37 | 0.72 |
| Item 9 | 0.57 | 0.70 |
| Item 10 | 0.40 | 0.72 |
| Item 11 | 0.43 | 0.72 |
| Item 12 | 0.52 | 0.71 |
| Item 13 | 0.43 | 0.72 |
| Item 14i | 0.35 | 0.72 |
| Item 15 | 0.37 | 0.72 |

**Note:**
Item numbers refer to the version adapted by *Modgill et al. (2014)*. "i" refers to items that have been reverse coded for scoring.

**Table 3 Means, standard deviations and distribution of responses to items of the OMS-HC scale.**

| Item | | | Responses | | |
| --- | --- | --- | --- | --- | --- |
| | M | SD | Agree | Neutral | Disagree |
| Attitudes of health care providers toward people with mental illness | 11.60 | 3.28 | | | |
| 1. I am more comfortable helping a person who has a physical illness than I am helping a person who has a mental illness. | 2.55 | 1.07 | 17.8% | 35.6% | 46.6% |
| 9. Despite my professional beliefs, I have negative reactions towards people who have mental illness. | 1.87 | 0.91 | 5.2% | 17.3% | 77.5% |
| 10. There is little I can do to help people with mental illness. | 1.83 | 0.99 | 8.7% | 12.6% | 78.7% |
| 11. More than half of people with mental illness don't try hard enough to get better. | 1.86 | 0.88 | 4.9% | 17.1% | 78% |
| 13. Health care providers do not need to be advocates for people with mental illness. | 1.73 | 0.89 | 5% | 12.6% | 82.4% |
| 15. I struggle to feel compassion for a person with a mental illness. | 1.77 | 0.88 | 4% | 14.2% | 81.8% |
| Disclosure/help-seeking | 9.33 | 2.75 | | | |
| 3. If I were under treatment for a mental illness, I would not disclose this to any of my colleagues. | 3.07 | 1.17 | 42.3% | 21% | 36.7% |
| 4. I would see myself as weak if I had a mental illness and could not fix it myself. | 2.54 | 1.25 | 29.5% | 14.2% | 56.3% |
| 5. I would be reluctant to seek help if I had a mental illness. | 1.56 | 0.86 | 5.1% | 3.6% | 91.3% |
| 8i. If I had a mental illness, I would tell my friends. | 2.16 | 1.01 | 72.5% | 16% | 11.5% |
| Social Distance | 9.86 | 3.05 | | | |
| 2i. If a colleague with whom I work told me they had a managed mental illness, I would be as willing to work with him/her. | 1.69 | 0.90 | 86.7% | 8.6% | 4.6% |
| 6i. Employers should hire a person with a managed mental illness if he/she is the best person for the job. | 1.78 | 0.92 | 81.5% | 14% | 4.5% |
| 7i. I would still go to a physician if I knew that the physician had been treated for a mental illness. | 2.13 | 0.98 | 69.2% | 21% | 9.7% |
| 12. I would not want a person with a mental illness, even if it were appropriately managed, to work with children. | 2.34 | 1.06 | 14.2% | 28.1% | 57.7% |
| 14i. I would not mind if a person with a mental illness lived next door to me. | 1.93 | 1.05 | 77.3% | 14.2% | 8.4% |
| OMS-HC total | 30.80 | 6.77 | | | |

**Note:**
Item numbers refer to the version adapted by *Modgill et al. (2014)*. "i" refers to items that have been reverse coded for scoring. Text presented here is the original English text. Participants answered the items in Spanish from the cross-culturally adapted version of the scale.

a positive attitude toward people with mental disorders. The total possible score varies from 15 to 75, and the total for the sample ($M = 30.80$, $SD = 6.77$), as well as the dimensions of secrecy/help-seeking ($M = 9.33$, $SD = 2.75$), attitudes of healthcare providers toward those with mental illness ($M = 11.60$, $SD = 3.28$), and social distance ($M = 9.86$, $SD = 3.05$), showed a tendency toward low levels of stigma. One example can be seen in the responses to the item "I would be reluctant to seek help if I had a mental illness," which showed the greatest percentage of disagreement (91.3%). However, there was a tendency to keep mental illness a secret, as seen in responses to the item "If I were under treatment for a mental illness, I would not disclose this to any of my colleagues," with 42.3% in agreement.

Table 4 presents the relationship between the sociodemographic variables and the total score of the OMS-HC. The results indicate that gender and occupation did not display any significant differences ($p = 0.897$ and $p = 0.203$, respectively), while discipline did, with a small effect size ($p < 0.01$, $f = 0.20$). The medicine and nursing groups had the highest levels

**Table 4 Sociodemographic variables and their relationship with the OMS-HC.**

| Variables | M | SD | F | df | p-value | ES |
|---|---|---|---|---|---|---|
| Gender | | | | | | |
|   Female | 30.81 | 6.91 | 2.678 | 1,554 | 0.897 | 0.014 |
|   Male | 30.72 | 6.20 | | | | |
| Discipline | | | | | | |
|   Medicine | 30.62 | 6.82 | 7.687 | 3,552 | <0.01** | 0.200 |
|   Nursing | 32.61 | 6.64 | | | | |
|   Clinical psychology | 27.95 | 5.77 | | | | |
|   Others | 32.04 | 6.76 | | | | |
| Educational level | | | | | | |
|   Technical education | 33.95 | 6.36 | 3.390 | 3,552 | 0.018* | 0.134 |
|   Bachelor's degree | 30.95 | 6.78 | | | | |
|   Master's degree | 29.25 | 6.63 | | | | |
|   Doctoral degree | 29.10 | 5.93 | | | | |
| Occupation | | | | | | |
|   Student | 30.22 | 6.59 | 1.601 | 2,553 | 0.203 | 0.075 |
|   Professional | 31.25 | 6.84 | | | | |
|   Both | 31.26 | 7.07 | | | | |
| Academic semester[a] | | | | | | |
|   1°–4° semester | 32.64 | 6.48 | 5.747 | 6,325 | <0.001** | 0.309 |
|   5°–6° semester | 34.09 | 5.25 | | | | |
|   7°–8° semester | 30.68 | 6.51 | | | | |
|   9°–10° semester | 27.80 | 6.68 | | | | |
|   Social service | 30.19 | 6.10 | | | | |
|   Specialization | 31.14 | 7.43 | | | | |
|   Graduate program | 27.02 | 5.58 | | | | |

**Notes:**
[a] Only "student" and "both" categories were included ($n = 332$).
* The correlation is significant at the 0.05 level (bilateral).
** The correlation is significant at the 0.01 level (bilateral). Results were confirmed by the nonparametric Mann-Whitney and Kruskal-Wallis U-tests.

of stigma, with significant differences between them (95% CI [−3.857 to −0.114], $p$-Tukey = 0.036), as well as between medicine and clinical psychology (95% CI [0.421 to −4.913], $p$-Tukey = 0.012), nursing and clinical psychology (95% CI [2.055–7.251], $p$-Tukey ≤ 0.001), and clinical psychology and other disciplines (95% CI [−7.380 to −0.796], $p$-Tukey = 0.008).

Significant differences were found in the educational level variable between groups, with a small effect size ($p = 0.018$, $f = 0.134$), particularly between master's degrees and technical education (95% CI [0.597–8.797], $p$-Tukey = 0.017), with the latter showing the highest levels of stigmatization. When the sample was restricted to students and those who were both students and professionals ($n = 332$) and analyzed by academic semester, the lowest levels of stigmatization were observed among those who were most academically advanced. Significant differences were found with a medium effect size ($p ≤ 0.001$, $d = 0.309$),

specifically between 1st–4th semester and graduate students (95% CI [0.035–11.206], $p$-Tukey = 0.047), between 5th-6th semester and 9th-10th semester students (95% CI [2.109–10.473], $p$-Tukey = ≤0.001), between 5th–6th semester and social service students (95% CI [0.056–7.739], $p$-Tukey = 0.044), between 5th–6th semester and graduate students (95% CI [2.825–11.304], $p$-Tukey ≤ 0.001), and between specialties and graduate students (95% CI [0.328–7.902], $p$-Tukey = 0.023). No correlation was found between age and the OMS-HC score (rho = 0.072, $p$ = 0.092).

## DISCUSSION

The findings of this study show that the OMS-HC scale has adequate psychometric properties for the evaluation of stigmatizing attitudes toward mental illness among healthcare personnel in Mexico. The measures to assess the fit of the model were adequate (*Mulaik et al., 1989*; *Hu & Bentler, 1999*; *Schermelleh-Engel, Moosbrugger & Müller, 2003*) and the three subscales identified correspond to the factorial structure proposed by *Modgill et al. (2014)*. Additionally, the global reliability of the scale (α = 0.73) was similar to that obtained in other adaptations. For example, in Singapore, α was found to be 0.75 (*Chang et al., 2017*), in Canada it was 0.77 (*van der Maas et al., 2018*), in Chile it was 0.69 (*Sapag et al., 2019*), in Hungary it was 0.73 (*Őri et al., 2020*), and in Germany it was 0.74 (*Zuaboni et al., 2021*). The subscales presented an internal consistency greater than or equal to 0.60 except for secrecy/help-seeking, which is consistent with previous studies (*Chang et al., 2017*; *Sapag et al., 2019*; *Zuaboni et al., 2021*). However, these values were less than the criterion of 0.70 for both coefficients, except for the omega value for the subscale Attitudes of Healthcare Providers Toward People with Mental Illness (ω = 0.70). *Tavakol & Dennick (2011)* have noted that subscales with few items tend to have low Cronbach's alpha values, suggesting that the secrecy/help-seeking and social distance subscale components might require a higher level of theoretical development. The internal consistency evaluation showed that all items significantly contributed to the scale. Moreover, it was observed that the elimination of any item does not produce an increase in the global value of the scale.

The administration of the OMS-HC to healthcare providers showed a general mean score of 30.80 (*SD* = 6.77) among the 556 participants in the sample. Given that the minimum score of the scale is 15 points and the maximum 75, the result is consistent with international studies conducted in Singapore (*M* = 35.7, *SD* = 6.4; *Chang et al., 2017*), Canada (*M* = 30.38, *SD* = 6.72; *van der Maas et al., 2018*), and Chile (*M* = 34.55, *SD* = 7.02; *Sapag et al., 2019*). While development of locally adapted measurements that consider the cultural context is recommended, this evidence of the validity of the OMS-HC in the Mexican population allows for cross-regional comparative studies (*Yang et al., 2007*; *Yang et al., 2014*). Overall, the findings suggest that stigmatizing attitudes among healthcare providers in the sample are comparable to those in other settings. Therefore, future research should explore the similarities and differences in these attitudes across different cultures and sociodemographic factors, to identify additional variables that could be associated with stigma.

The comparison of the mean scores of the OMS-HC with sociodemographic characteristics found no significant relationships between stigmatization levels and age or gender variables. This is consistent with previous research using the same scale (*Chang et al., 2017*; *Destrebecq et al., 2018*; *Sapag et al., 2019*), suggesting that these variables alone do not seem to be related to stigma. However, these variables may be related to other conditions, such as education, personal experience, and mental health literacy.
No significant differences were found in this sample regarding the relationship between stigma and occupation (student, professional, or both). This finding would seem to reinforce the results of previous studies by *Kopera et al. (2015)* and *Carrara et al. (2019)*, suggesting that everyday contact does not necessarily modify negative attitudes toward those with mental health disorders. Although professionals have more frequent contact with these individuals than students, the quality of social interactions may be negatively impacted by factors such as organizational culture, structural stigma, and work overload, as suggested by *Henderson et al. (2014)*. Therefore, it is essential to consider how these external conditions may influence the stigma reduction process.

The results do, however, indicate that discipline has a certain effect on stigma levels. Nursing and medical providers had higher stigmatization scores than psychology staff, which is consistent with other studies (*Chang et al., 2017*; *Sapag et al., 2019*). *Lauber et al. (2006)* suggest that professional background may have a slight influence on perpetuating negative stereotypes, whereas *Cleary & Dowling (2009)* note that differences in stigmatization levels may be due to variations in the role and responsibilities of healthcare providers in treating individuals with mental health disorders, such as familiarity with the recovery process, the importance of therapeutic risk, symptom management, and the causes of mental illness.

An association was found between educational level and stigmatization. Although the effect size is moderate, it was observed that the level of stigma decreases as educational level increases. This trend was also observed in the student subsample, with stigma scores being lower in later than early semesters, where a medium effect was observed. According to *Evans-Lacko et al. (2010)*, the presence of certain types of knowledge could contribute to the reduction of stigmatization, especially those associated with symptom recognition and the diversity of effective treatments. This could also be related to a higher level of experience and quality of contact during clinical practice (*Henderson et al., 2014*).

The findings of the present study point to the need to create specially designed interventions to reduce the stigmatization of mental illness by healthcare providers at various levels of care. Attitudinal factors, particularly those related to social contact, are one of the main components for the reduction of stigmatization in this group (*Stuber et al., 2014*). It is therefore necessary to study the effects of stigmatization on and between different contexts.

## Limitations of the study

First, it is important to note that convenience non-probability sampling was used, which limits the generalization of the findings to other population groups. This study evaluated factorial validity and internal consistency, but it is also essential to carry out

criterion-related validity studies with other stigmatization scales, including scales of mental health literacy and intentions to discriminate. Second, stigma-related issues can generate biases of social desirability, which could have led to low scores on the OMS-HC. However, this limitation may have been mitigated by the self-report format in which the questionnaires were administered, in addition to the measures that allowed participants to respond anonymously. Third, certain contact-related variables, such as regular experience with mental health patients or having had a mental health problem themselves or a family member who did, could be determinants for the development of certain stigmatizing attitudes. However, these variables were not included in the study and should be considered in future research for a better understanding of stigma and discrimination in this context.

Finally, the fact that the COVID-19 pandemic required two different methods of data collection may have had an influence on the responses obtained, even though the same guarantees of confidentiality were provided.

## CONCLUSIONS

The Spanish adaptation for Mexico of the OMS-HC scale is a tool that shows good psychometric properties for measuring the stigmatization of mental illness among healthcare providers. The Spanish adaptation of the scale will enable cross-cultural and cross-disciplinary comparisons, as well as evaluation of the effectiveness of interventions designed to reduce stigmatizing attitudes. Although it is important to emphasize that two of the subscales showed omega values below the acceptable level, the reliability of the scale measured by the total score was acceptable ($\alpha = 0.73$, $\omega = 0.76$). The use of the total score seems more suitable than the subscale scores and we suggest further research into the factorial structure of the instrument. The findings of this study reveal the presence of stigmatizing attitudes in the Mexican population. Therefore, targeted interventions in the healthcare sector at different levels of care are necessary to address this issue. As healthcare providers are often the first point of contact for individuals with mental health disorders, research on stigmatizing attitudes toward mental health among healthcare providers in Latin America is urgently needed.

## ACKNOWLEDGEMENTS

We thank Scott Patten and his team for sharing the scale for its adaptation to Spanish. Finally, we would like to thank the participants for their time, dedication, and collaboration in this study.

### Funding

The Mexican National Council on Science and Technology (CONACyT) awarded a fellowship to Hugo Noel Valdivia Ramos (No. CVU 642488). The funders had no role in study design, data collection and analysis, decision to publish, or preparation of the manuscript.

## Grant Disclosures

The following grant information was disclosed by the authors:

The Mexican National Council on Science and Technology (CONACyT): CVU 642488.

## Competing Interests

The authors declare that they have no competing interests.

## Author Contributions

- Hugo Noel Valdivia Ramos conceived and designed the experiments, performed the experiments, analyzed the data, prepared figures and/or tables, authored or reviewed drafts of the article, and approved the final draft.
- Jazmín Mora-Rios conceived and designed the experiments, performed the experiments, analyzed the data, authored or reviewed drafts of the article, and approved the final draft.
- Guillermina Natera conceived and designed the experiments, authored or reviewed drafts of the article, and approved the final draft.
- Liliana Mondragón conceived and designed the experiments, authored or reviewed drafts of the article, and approved the final draft.

## Human Ethics

The following information was supplied relating to ethical approvals (*i.e.*, approving body and any reference numbers):

Comisión de ética de la Universidad Nacional Autónoma de México.

## Data Availability

The raw data is available in the Supplemental File.

## Supplemental Information

Supplemental information for this article can be found online at http://dx.doi.org/10.7717/peerj.16375#supplemental-information.

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
