# Peer review of "Psychometric properties of the Mexican version of the opening minds stigma scale for health care providers (OMS-HC)"

_PeerJ, doi:10.7717/peerj.16375_

## Round 0.1 · original submission · Major Revisions

I believe that all comments from the two reviewers are very helpful for your manuscript. So please see their comments and revise your manuscript.

Reviewer 1 ·

Basic reporting

I commend the authors the manuscript is cleary written in professional, unambiguous language.
Structure conform to Peerj standars
Figures are relevant, high quality, but don't forget the title in figures.
Raw data supplied (I suggest send in a excel file)

Your introduction needs more detail. I suggest that you improve the description at lines 65- 74 to provide more justification for your study (specifically, you should expand upon the knowledge gap being filled).
In Latin America, instruments for assessing stigma are scarce. Therefore, the authors should update the information by considering the latest reports on this matter. A literature search yielded the following references:
Sapag, J. C., Klabunde, R., Villarroel, L., Velasco, P. R., Álvarez, C., Parra, C., ... & Corrigan, P. (2019). Validation of the Opening Minds Scale and patterns of stigma in Chilean primary health care. PloS one, 14(9), e0221825.
Vielma-Aguilera, A., Bustos, C., Saldivia, S., & Grandón, P. (2023). Psychometric properties of the attitudes scale of health care professionals toward people with a diagnosis of mental illness (EAPS-TM). Current Psychology, 42(7), 5851-5863.
Bolivar-Paredes, E., & Villanueva-Ruska, A. (2017). Validation and reliability of the AQ-27 Questionnaire on stigmatizing attitudes towards patients with schizophrenia in a General Hospital.

Experimental design

In the Materials and Methods section, lines 112 to 117, the authors do not clearly explain why the data collection was conducted in two stages. It can be inferred that the change in procedure was related to the COVID-19 situation. How do the authors justify this change in data collection procedure? Could it have any implications for the results as well as the quality of the obtained information? Additionally, this information is repeated in the Procedures section.
In the Participants section, the authors should provide a breakdown of how many participants are students and how many are professionals.
In the Instruments section, a corresponding citation should be inserted at line 138.
In the Data Analysis section, lines 204-205, it is recommendable to include McDonald's Omega coefficient (McDonald, R. P. 1999. Test theory: A unified treatment. Mahwah: Lawrence Erlbaum Associates, Inc.).

Validity of the findings

All underlying data have been provided; they are robust, statistically sound, & controlled.

Shang et al., 2017 (BMJ Open 2017;7:e018099. doi:10.1136/bmjopen-2017-018099, página 2) and Sapag et al., 2019 (PLoS ONE14(9): e0221825. https://doi.org/10.1371/journal.pone.0221825, página 2) provide contextual arguments for adapting the WHO-HC scale. What are the arguments for the adaptation in the Mexican population?.

·

Basic reporting

Thank you very much for the opportunity to review this manuscript.
This is an article that attempts to test the factor structure of the OMS-HC in Mexico, which has already been investigated in other countries.

The manuscript is overall clear; it is easy to follow; however, there are some inaccurate terms. This text might benefit from proofreading service as the English language could be improved in several parts of the manuscript - for example "further validation by Modgill," (inaccurate term), stigmatizing toward mental health etc. to ensure that an international audience can clearly understand the text and that the appropriate terms are used.

The authors provide sufficient background to their study. The only thing that I miss from this manuscript are more detail from other Spanish versions of the OMS-HC.
As Spanish language versions of the scale were adapted in Chile and Spain, I suggest writing in more detail, including fit indices and findings from all other Spanish OMS-HC studies: Sapag 2019 and Ori et al. 2023.
(Sapag 2019: https://journals.plos.org/plosone/article?id=10.1371/journal.pone.0221825
(Ori 2023: https://www.frontiersin.org/articles/10.3389/fpubh.2023.1168929/full)
Note that Sapag omitted items - for example.

The article structure, figures, and tables are clear. (Please find below what I missed from the sociodemographic table).

Experimental design

As the OMS-HC has not yet been investigated in Mexico, it is important to have a Mexican version of the scale.

Ethical standard is appropriate, and it is also important that the research group requested permission to use the scale.
The measures used in this study appear to be appropriate for the research question and study
design.
However, It is important to note that the sample size and participant demographics are not clearly reported: the discipline was the medicine of 59% of the sample (n=328). What was their specialty? It is very important to know how homogenous the sample was. Moreover, there can be differences in the stigmatizing attitudes of different professionals. Please provide more information.

Please provide more detail on how you handled with missing data.

The statistical methods used to analyze the data are appropriate and well-described. However, it should be highlighted that this is not a whole validation process of the Oms-HC for Mexico as it lacks several parts of the proper validation procedure. Please check other forms of reliability and validity:
Arafat SY, Chowdhury HR, Qusar MM, Hafez MA. Cross-cultural adaptation and
psychometric validation of research instruments: A methodological review. J Behav Health.
2016 Jul 14;5(3):129-36. DOI: 10.5455/jbh.20160615121755

It is fine that you included this issue in the limitation section. However, it is important to erase and rewrite all parts that state that the OMS-HC is a valid and reliable measurement of stigma in Mexico.

Validity of the findings

The strongest part of the study is the clearly written data analysis of the method section and results section. It is also appreciated that effect sizes are provided.
Please consider the above-mentioned issue that this research cannot be considered as a whole validation procedure of the scale. Please rewrite the conclusion accordingly.

Additional comments

I commented on the manuscript as well.
Overall, this study provides valuable insights into the psychometric properties of the OMS-HC
scale in Mexico. The manuscript needs some refinement; therefore, I suggest publication after major revision.

---

## Round 0.2 · Minor Revisions

Dear Dr. Valdivia Ramos,

Thank you for submitting your manuscript to PeerJ. The original Academic Editor is not available and so I have taken over handling the submission.

We have just obtained the reviews from our experts. I also have read the manuscript myself independently before looking at the reviews. Overall, we are satisfied with the revision and agree that your paper has potential for an impactful contribution.

The reviewers also raised a number of relatively minor concerns that should be easy for you to address. Please address their comments thoroughly. Pending the revision, I am happy to conditionally accept your paper for publication in PeerJ

Thank you for giving us the opportunity to consider your work, and we wish you all the best with your article.

Yours sincerely,

Andree Hartanto
Academic Editor
PeerJ

Reviewer 1 ·

Basic reporting

The authors have addressed the concerns raised in the initial review, resulting in an improved presentation of the document. However, I have a observation:

Regarding the previously mentioned question, "What is the rationale for conducting an adaptation of the same instrument in Mexico?", the argument still appears weak.

Experimental design

In the procedures section, it is advisable to explicitly mention the process of translation and back-translation, as suggested by the authors cited in the document (Sousa & Rojjanasrirat, 2011), or alternatively, outline the steps proposed by those authors.

Validity of the findings

no comment

Additional comments

Procedure section:
Results and references to the Spanish versions from Spain (Öri et al., 2023) and Chile Sapag, et al., 2019) are presented. The Chilean version demonstrates a three-dimensional structure similar to Modgill's version (Öri et al., 2023).
Why didn't you use the Chilean adaptation and instead decide to translate Modgill's version?

·

Basic reporting

I commend the authors for all their work on the revision, which I believe resulted in a significant improvement in the quality of the manuscript. I would recommend the following issues to consider when finalizing the manuscript for publication.

The inclusion of a native English speaker editor was an excellent choice. The introduction still contains some unusual-sounding text. For example: "Originally consisting of items, it was adjusted to a two-factor structure with twelve items."

Experimental design

Lines 263-265: Please provide the guides for alpha and omega.

Validity of the findings

It is pleasant that the authors included the omega coefficient as it is a more accurate reliability measurement and has several advantages compared to Cronbach’s alpha coefficient. This is not an easy area of research as the cut-off scores for omega are not cast in stone, but according to the generally accepted guides, >0.7 indicates acceptable reliability(McNeish D. Thanks coefficient alpha, we'll take it from here. Psychol Methods. 2018 Sep;23(3):412-433. doi: 10.1037/met0000144. Epub 2017 May 29. PMID: 28557467, https://pubmed.ncbi.nlm.nih.gov/28557467/).
For this reason, it appears that the reliability of the scale itself and one subscale are in the acceptable range, but the remaining two subscales appear to have lower-than-acceptable reliability.
Thus, the Line 371: “The values of McDonald’s omega were adequate in all cases.“ should be adjusted and modified accordingly, drawing attention to the fact that two subscales had lower than acceptable omega coefficients.

The sentence should also be modified to make it clear to the future readers that the coefficients were acceptable for the scale (measured by the total score) and is not necessarily true for the three subscales.

This should also be included in the conclusion section.

Please also indicate this in the abstract.
For example,
The model demonstrated good fit (x2/df = 2.36, RMSEA = 0.050, CFI = 0.970, TLI = 0.962, SRMR = 0.054, NFI = 0.95950, PNFI = 0.742). Internal consistency was found to be adequate (α = 0.73, ω = 0.76) for the scale itself and lower than/slightly lower than acceptable for the subscales.

Additional comments

Minor comments:
Line 162 the year is missing. (I know it is given above, however, to improve clarity please add it with the months.)
Line 179, inconsistent data reporting. Only percentages are given, subsample sizes are missing
Line 183: “…although we did not collect data on their field of study” seems to be more accurate.

---

## Round 0.3 · accepted · Accept

Dear Dr. Valdivia Ramos,

I am pleased to advise that the above paper has now been accepted for publication in PeerJ. Thank you for giving the Journal the opportunity to publish your work. We are impressed with your paper and believe that it will contribute well to the literature. Well done!

In addition, Reviewer 2 has made a constructive suggestion and I hope that you can revise the sentence as suggested when you receive your proofing PDF.

Best Regards,
Andree

Reviewer 1 ·

Basic reporting

After conducting a comprehensive examination, it becomes evident that the authors have implemented the suggested modifications.. Their arguments are appropriately presented. Furthermore, they have incorporated pertinent bibliographic references. I recommend a thorough review of the formatting of the bibliographic citations within the manuscript, as they appear to adhere to the Vancouver style (arranged in order of appearance) rather than the conventional alphabetical order.

Experimental design

no comment

Validity of the findings

no comment

Additional comments

no comment

·

Basic reporting

fair

Experimental design

-

Validity of the findings

-

Additional comments

I appreciate the authors' additional efforts toward improving their manuscript. I suggest correcting the issue below. After this minor correction, I believe it will be ready for publication.

Line 417: "...the total reliability of the scale was acceptable ( = 0.73, = 0.76). For this reason, we suggest further research into the factorial structure of the instrument."

The term total reliability does not make any sense and the results indicate the reliability of the total score and the weakness of the subscales, which can be mentioned.

For example: "....the reliability of the scale measured by the total score was acceptable (= 0.73, = 0.76). The use of the total score seems more suitable than the subscale scores and we suggest further research....".